# Blastfurnace Hybrid Cement with Waste Water Glass Activator: Alkali–Silica Reaction Study

**DOI:** 10.3390/ma13163646

**Published:** 2020-08-17

**Authors:** Lukáš Kalina, Vlastimil Bílek, Lada Bradová, Libor Topolář

**Affiliations:** 1Faculty of Chemistry, Brno University of Technology, 61200 Brno, Czech Republic; bilek@fch.vut.cz (V.B.J.); lada.bradova@vut.cz (L.B.); 2Faculty of Civil Engineering, Brno University of Technology, 60200 Brno, Czech Republic; libor.topolar@vutbr.cz

**Keywords:** hybrid cement, alkaline activation, blast furnace slag, waste water glass, ordinary Portland cement, alkali–silica reaction, aggregate

## Abstract

Hybrid systems represent a new sustainable type of cement combining the properties of ordinary Portland cement and alkali-activated materials. In this study, a hybrid system based on blast furnace slag and Portland clinker was investigated. The economic aspects and appropriate waste management resulted in the usage of technological waste from water glass production (WG-waste) as an alkaline activator. Although the Portland clinker content was very low, the incorporation of this by-product significantly improved the mechanical properties. Nevertheless, the high amount of alkalis in combination with possible reactive aggregates raises concerns about the risk of alkali–silica reaction (ASR). The results obtained from expansion measurement, the uranyl acetate fluorescence method, and microstructure characterization revealed that the undesirable effects of alkali–silica reaction in mortars based on the hydration of hybrid cement are minimal.

## 1. Introduction

It is well known that cement is one of the most used construction materials in the world. Nowadays, global cement production has greatly increased due to the rising population, continual urbanisation, and infrastructure development [1]. For this reason, cement consumption in 2018 was 4.1 billion tons [2], which is unfortunately associated with a significant demand for energy and CO_2_ emissions.

The global approach to reducing the carbon footprint of cement contains several options. Firstly, alternative fuels and/or raw materials could be used. Further, the replacement of Portland clinker with supplementary cementitious materials (SCMs) is an option. The development of alternative binders not based on Portland clinker represents the third option, and finally, the capture of CO_2_ emitted by cement plants could be another solution [3]. In most parts of the world, the utilization of SCMs is particularly advantageous, especially if “low-carbon” materials such as industrial waste products are used. Blast furnace slag (BFS), a by-product from pig iron production, or fly ash from coal combustion provide a viable way to substitute Portland clinker [4]. On the other hand, these blended cements with a high content of SCMs can reduce early strength development, which represents a considerable drawback in practical applications [5]. Despite the low clinker content, one of the potential solutions is the production of new cement binders, such as hybrid cements. Hybrid cements aim to combine the positive characteristics of traditional ordinary Portland cement (OPC) binders with those resulting from the alkaline activation process, generating materials with high application potential. The production of hybrid binders can be achieved in two different approaches. The first option is the activation of aluminosilicate materials (mixture of OPC and SCMs) with high alkaline solution which is the conventional procedure for alkali-activated material preparation. The second possibility consists of the production of a special type of cement via incorporation of an alkaline activator mixed in solid form together with other cement components, such as BFS and Portland clinker. The binder formed in this way provides two major benefits. It can be activated only with water, and subjected to certain criteria, it can be classified as Blastfurnace cement in accordance with European standard EN 197-1. This study is focused on this non-traditional type of cement.

The promising behaviour of hybrid binders in terms of mechanical properties has been reported in several studies [6,7,8]. However, the other aspects related especially to the durability are not yet well documented. The highly alkaline character of hybrid systems creates concern about the alkali–silica reaction (ASR), the reaction between alkalis from the alkaline activator and reactive silica present in the coarse or fine aggregates. In alkali-activated materials (AAMs), the extent of ASR is related to the character of the raw materials and the alkaline activator being used. Chen et al. [9] indicated that the highest expansion exhibited water glass-activated BFS cements in comparison with BFS cements activated by other known activators such as NaOH, Na_2_CO_3_ or Na_2_SO_4_. Regardless of the selection of activator, there is some evidence that the risk of ASR increases with a higher amount of alkali dose [10] and the basicity of BFS [9]. Nevertheless, a comparison with OPC-based materials suggests that the expansion caused by ASR is lower in AAMs (by approximately 50 %) [11,12] which is probably related to the conditions of its origin that are further discussed.

In general, the ASR is a set of complex physical–chemical processes that give rise to the formation of expansive alkali–silica gel causing the deterioration of concrete. The explanation of the ASR mechanism has been an issue over the years; nevertheless, recent studies [13,14] assume that a key role is played by Ca^2+^ ions and the alkali–silica gel creating a rigid rim that tightly surrounds aggregate particles. The rigid reaction rim acts as a semi-permeable membrane that allows the transfer of alkali, Ca^2+^, and OH^–^ ions, which cause the ASR of the aggregate to be produced. The result is that the internal pressure is stored in the aggregate and its accumulation subsequently leads to crack formation. Although there are chemical admixtures capable of ASR reduction, the positive effect of SCMs has been verified [15]. These mineral admixtures have the ability to reduce ASR because of two main reasons. Firstly, during the hydration process in the presence of SCMs, the alkali content in the pore solution is decreased due to the formation of C-S-H gel with a low Ca/Si ratio able to bind alkaline ions in its structure [16]. Secondly, the concentration of Ca^2+^ ions in the pore solution is also reduced thanks to the reaction of Ca(OH)_2_ and the amorphous phases of SCMs which suppress the formation of the rigid rim around the aggregate particles [13].

The studies concerning ASR in AAMs report a lower expansion rate compared to binders based on OPC. However, the incorporation of alkalis into OPC systems to form hybrid systems represents a new type of one-part alkali-activated material where the consequences of ASR can be an issue. Therefore, the aim of this work was to evaluate the potential risk of ASR in highly alkaline blastfurnace hybrid cement. At the same time, the beneficial effect of the alkali content on the hydration process in this cement was demonstrated. Moreover, the designed hybrid system exhibits considerable ecological importance in terms of water glass waste processing, which is currently landfilled. The comparison of obtained results using the commercial blastfurnace CEM III/B and CEM I cement promises the possibility of the utilization of a new sustainable type of cement with a very low Portland clinker content.

## 2. Materials and Methods

### 2.1. Materials

A hybrid blastfurnace cement (CEM III/C-H) was designed in accordance with the EN 197-1 standard for Blastfurnace CEM III/C cements with respect to the required chemical and physical properties and was prepared by milling together Portland clinker—5 wt. % (Cemmac, Horné Srnie, Ltd., Slovakia), blast furnace slag—more than 80 wt. % (ArcellorMittal Ostrava, Ltd., Czech Republic) and dried technological waste from water glass production (Vodní sklo, Inc., Brno, Czech Republic). The phase composition of used materials was determined using an X-Ray powder diffraction (XRD) analyser EMPYREAN (PANalytical, Almelo, Netherland) in a central focusing arrangement using CuKα radiation with step 0.013 °2θ. The method of the internal standard (calcium fluorite) for the amorphous part determination was applied. The main minerals identified in BFS were calcite (6.7%), akermanite (19.8%), merwinite (0.4%), and quartz (2.7%). The XRD analysis showed the content of an amorphous phase of about 70.5%. Portland clinker contains common phases such as dicalcium silicate (9.3%), tricalcium silicate (67.7%), tricalcium aluminate (2.5%), tetracalcium aluminoferrite (10.8%), and lime (0.5%). The amount of the amorphous phase was measured to be 9.3%. The water glass waste (WG-waste) contained 7.89 wt. % of Na_2_O and 16.03 wt. % of SiO_2_ in soluble fraction determined by conductometric titration. The insoluble part of WG-waste is represented by siliceous sand. The WG-waste was added to the hybrid cement with a given Na_2_O/BFS weight ratio of 1.2. The specific surface area of CEM III/C-H was 580 m^2^/kg. The comparative cements CEM I (390 m^2^/kg) and CEM III/B (480 m^2^/kg) were obtained from Horné Srnie cement plant (Cemmac, Ltd., Slovakia). The chemical compositions of used cements are summarized in Table 1.

Natural crushed aggregate (CA) from Luleč quarry (HeidelbergCement Group, Mokrá, Czech Republic) and CEN standard siliceous sand (SS) meeting the requirements of EN 196-1 were used for verification of ASR performance of cement samples. Both aggregates were indicated to be deleterious according to ASTM C289-07. The mineralogical composition of crushed aggregates showed phases such as quartz (41.8%), albite (25.4%), orthoclase (16.5%), and anorthite (14.6%), with an amorphous phase of 1.7%. The standard siliceous sand was composed of quartz with a minimum content of the amorphous phase (below 0.5%).

### 2.2. Testing Methods

The method for determining the ASR potential of samples with various types of cement and aggregates was implemented based on ASTM C1567-13. The cured mortar bars (24 hrs. in molds under laboratory conditions; 24 hrs. in water at 80 °C) with specific grading requirements for aggregates were exposed to 1N NaOH solution at 80 °C for 14 days. During the 14-day test, the length of the mortar bars was recorded every day using an ASTM C490 apparatus (JIP-TECH, Ltd., Prague, Czech Republic). After the test, the expansion extent of samples was evaluated.

The preparation of mortar samples and compressive strengths of cement mortars were determined in accordance with EN 196-1 at 2, 7, and 28 days and were carried out by means of a compressive and bending strength tester Betonsystem Desttest 3310 (Betonsystem, Ltd., Brno, Czech Republic). Moreover, the ASR impact on compressive strength development was evaluated on 28-day-old samples exposed to the ASTM C1567-13 procedure.

The uranyl acetate fluorescence method was used for the identification of ASR on the cross-section of the mortar bars after the ASTM C1567-13 procedure was applied. Compared to the mentioned standard, only a fraction of the aggregate in the range of 1500 to 3000 μm was used for better observation of the ASR. Uranyl acetate solution was prepared by adding 5 g of uranyl acetate powder to 200 mL of a 0.4 N acetic acid solution. By applying uranyl acetate to the surface, the alkali–silica gel imparts a characteristic yellowish-green glow under UV light (254 nm) which was recorded by a Nikon D5100 camera (Nikon Corporation, Tokyo, Japan).

Scanning electron microscopy was used to examine the fracture surfaces in the backscattered electron mode using a JEOL JSM 7600F (JEOL, Ltd., Tokyo, Japan) electron microscope. The specimens were stuck onto carbon tape, and the exposed fractured surfaces were sputter coated with gold. The working distance in the measurement process was set to 12 mm, and the accelerate voltage was 15 kV. Energy dispersive X-ray results of the sample surface were obtained at selected locations using an Ultim^®^ Max 100 mm^2^ detector (Oxford Instruments, Plc., Abingdon, UK) and were evaluated with an Oxford AZtec System (Oxford Instruments, Plc., Abingdon, UK).

## 3. Results and Discussion

### 3.1. Physical–Mechanical Properties

The expansion measurement based on the accelerated expansion test (ASTM C1567-13) is shown in Figure 1. According to these results, both mortars with blastfurnace types of cement expand less than 0.10% at 16 days and therefore have a low risk of deleterious expansion when used in concrete under field conditions. The opposite situation occurs when CEM I type of cement is used with a combination of potential deleterious aggregates. The expansion is greater than 0.20% after 16 days which raises concerns over a low resistance to ASR. The obtained results suggest that a high level of clinker replacement for blast furnace slag in both the CEM III-B and hybrid system (CEM III/C-H) results in a considerable reduction of undesirable expansion. The high alkali content in the hybrid type of cement should be a critical factor determining the occurrence of ASR. Previous studies confirmed that an increasing content of alkalis in alkali-activated slag cement mortars causes ASR expansion, especially when water glass activators are used [9,10,17]. In this case, it can be seen that the contribution of blast furnace slag is a more crucial parameter than the high alkali content in hybrid cement caused by WG-waste addition. This finding is consistent with previous research where it was verified that supplementary cementitious materials such as BFS were very effective at eliminating damaging ASR expansion even at low replacement levels [15,18]. Therefore, hybrid cement mortars exhibit a similar behavior in the expansion development as mortars with CEM III/B cement. Moreover, when the standard siliceous sand was used as an aggregate, the greatest mitigation of expansion in the case of hybrid mortar bars was observed in the first days compared to samples prepared with other types of cement. It was also obvious that the development of expansion in the case of SS sand is more extensive in comparison with mortars prepared by CA aggregates. This is probably related to the chemical composition of aggregates. A higher content of SiO_2_ in SS sand causes a higher rate of dissolution of silica species into the pore solution; therefore, the possibility of creating ASR products is higher.

The extent of ASR also correlated with the compressive strength development (Figure 2). After the standard water-curing process, the samples were exposed to the conditions equal to the accelerated ASR mortar-bar method. The results indicate that the environment initiating ASR (1 N NaOH; 80 °C) affects the strength development of samples with blastfurnace types of cement rather favorably, and the risks of undesirable expansion are minimized. This phenomenon is probably related to the dissolution of not yet reacted slag particles and the formation of additional binder phases via the alkaline activation process. The opposite situation is shown in samples with CEM I cement. It is not surprising that the development of compressive strength under standard conditions is faster than for blastfurnace cement. However, due to initiated ASR, a significant decrease in strength was observed, resulting in poorer compressive strengths when compared with samples prepared with CEM III/B or hybrid cement. A difference in compressive strength development between the used aggregates was also observed. The mortar samples with SS sand exhibited mostly higher compressive strengths compared to samples with CA aggregates. It is well known that the shape and surface texture of aggregate particles particularly influence the properties of fresh concrete. The usage of smooth and rounded aggregate particles increases the workability of the mortar mixture which is then in a hardened state more uniform and less porous. Subsequently, one can expect that this aspect strongly affects compressive strength development.

### 3.2. Determination of ASR Using the Uranyl Acetate Method

The dye method with a uranyl acetate solution to detect ASR gel was applied after completion of the accelerated expansion test. This method was intended to reveal the ASR products potentially presenting near-aggregate grains. Figure 3 shows cross-section areas of mortar bars under ultraviolet light after the application of a uranyl acetate solution. It was observed that only the samples prepared with CEM I cement contain yellowish-green glow rims surrounding the aggregate grains originating from alkali substitution by uranyl ions in the ASR gel. Contrary to this, the absence of mentioned rims was not detected in the case of CEM III/B and CEM III/C-H samples, suggesting that the formation of an expanding gel is minimal. However, the cross-section areas of mortars with blastfurnace cement show a significant yellowish-green fluorescence of the binder phase itself. This is mainly related to the character of the created C-S-H gel. It is well known that during the hydration of OPC with BFS addition as well as in the alkaline activation process of BFS, the substitution of silicon by aluminum tetrahedra takes place which gives rise to the formation of C-A-S-H gel [19,20]. This replacement generates a charge imbalance in the structure which is compensated by positively charged ions, in particular by Ca^2+^ or alkali metal ions [21]. During the dye method, cation exchange with uranyl groups may occur, resulting in the bright green color of the binder phase under UV light. Nevertheless, it should be noted that the positively charged uranyl ions are also able to bind to the exposed BFS grains in the cross-section area. The silanol groups on the wetted surface of unreacted slag particles are deprotonated in the alkaline environment which induces a negatively charged surface [22], allowing the uranyl cations to be attracted to them via the electrostatic forces.

Finally, with respect to the used method, it is important to point out that despite the hybrid cement (CEM III/C-H) containing a considerable amount of alkalis and siliceous residues from WG-waste, the deleterious ASR gel was not found.

### 3.3. Microstructure Characterization

A comparison of hydration products located near the aggregates was performed on mortars showing significant volume changes (addition of CEM I cement) and volume-stable mortars based on CEM III/C-H cement. Figure 4 shows the fracture areas of both samples after the accelerated expansion test (ASTM C1567-13). In the case of hybrid systems, a uniform structure where the grains of aggregates were embedded in the binder phase (C-A-S-H gel) was observed. The determined chemical composition of the C-A-S-H gel was very similar to gels originating from the hydration process of blended cements (place 3, 4, 5; Table 2).

The mean Ca/Si and Al/Ca ratios were 1.39 ± 0.05 and 0.11 ± 0.03, respectively, which is typical for hydration products of cement with the high replacement of Portland clinker by BFS [23]. Simultaneously, no substantial changes in the chemical composition of the C-A-S-H gel were observed in gels near the aggregates and the matrix itself, and no products of ASR were detected in mortars with natural crushed aggregates or in standard siliceous sand. This fact is related mainly to the nature of the C-A-S-H gel. The low Ca/Si ratio of the gel has a high capacity to fix alkaline ions, and thus less free alkalis are available for the creation of the ASR gel [16]. A similar effect was also achieved with other SCMs such as silica fume [17], low-calcium fly ash [24], and metakaolin [25]. It follows that the concentration of Ca^2+^ ions in pore solution is a key parameter with regards to ASR development.

A different situation is shown in mortars with the CEM I type of cement. ASR products occur in the immediate vicinity of some aggregate grains. The character of these products varied considerably. Based on scanning electron microscopy (SEM) was detected both the rosette-type crystals and the amorphous gel near the siliceous aggregate grains (Figure 4). A previous study [26] suggested that the type of gel is related to its aging. While young gels are characterized by smooth textures, older gels often crystallize as rosette shapes. Additionally, the presence of calcium ions during the formation of the expansive gel plays an important role. It was verified that at a high Ca/Si ratio, ASR products are destabilized to amorphous C-S-H gels whereas crystalline ASR products are favorably formed at a low Ca/Si ratio under 0.5 [27]. That is consistent with the EDS results from the observed ASR products (place 1 and 2; Table 2). Whereas the rosette crystals had a low Ca/Si ratio of 0.44, the ratio in the amorphous phase near the standard siliceous sand was determined to be 1.12. In both cases, a high amount of alkali was detected.

## 4. Conclusions

The potential risk of ASR in mortars prepared using a new type of hybrid cement has been investigated. At the same time, a comparison with commercial blastfurnace CEM III/B and CEM I types of cement was carried out. Based on the obtained results, the following conclusions can be drawn:The designed composition of the hybrid cement showed very good resistance to ASR, while containing a high amount of alkalis and siliceous residues from WG-waste. The expansion during the accelerated test was very low (under 0.1%) and did not negatively affect the mechanical properties of the prepared mortar samples. The compressive strength development continued to increase even after exposure to 1N NaOH at 80 °C.Despite the fact that the mortars prepared from hybrid cement contained the deleterious types of aggregate, the ASR products were not detected in contrast with mortars based on CEM I cement.Microstructure characterization revealed the ASR products only in the case of mortars with CEM I cement. The chemical composition of the binder phase in hydrated hybrid cement did not show significant changes in gels near the aggregates and the matrix itself.The increased alkali content in hybrid cement did not lead to a deleterious ASR expansion, and simultaneously, the performance was practically the same as that of the CEM III/B cement. Therefore, a sufficient slag content seems to be a key parameter for maintaining very low ASR expansion.

## Figures and Tables

**Figure 1 materials-13-03646-f001:**
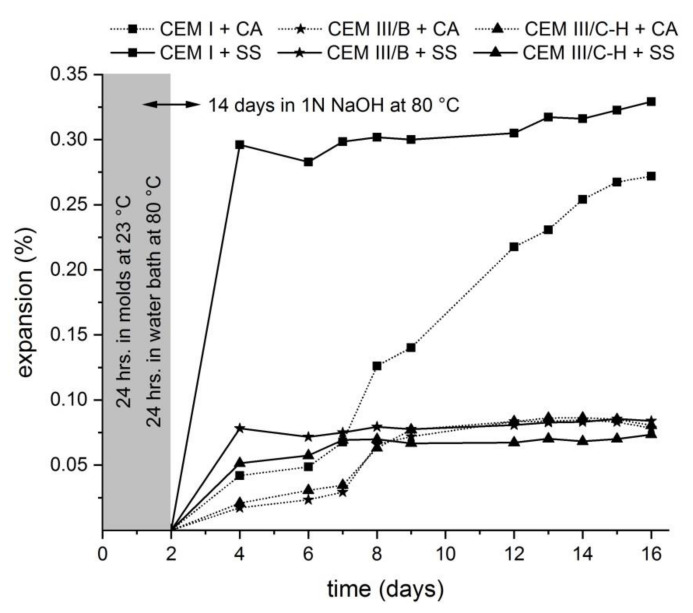
Length change of mortar bars under the condition given by ASTM C1567-13.

**Figure 2 materials-13-03646-f002:**
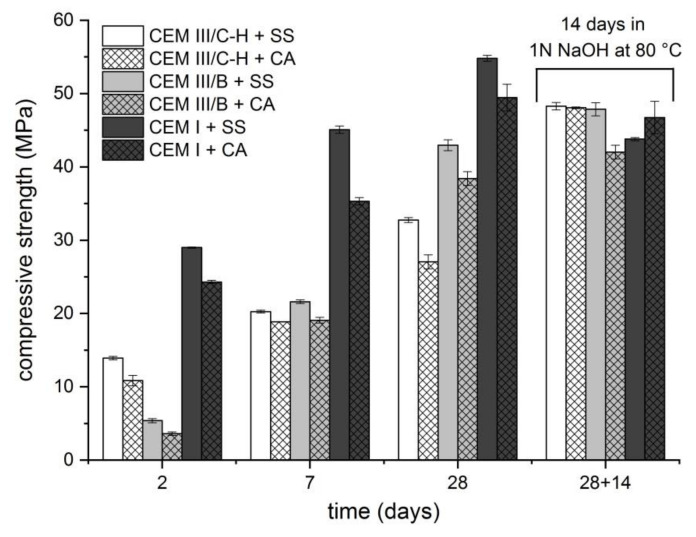
Compressive strength development of mortar bars prepared using different types of cement.

**Figure 3 materials-13-03646-f003:**
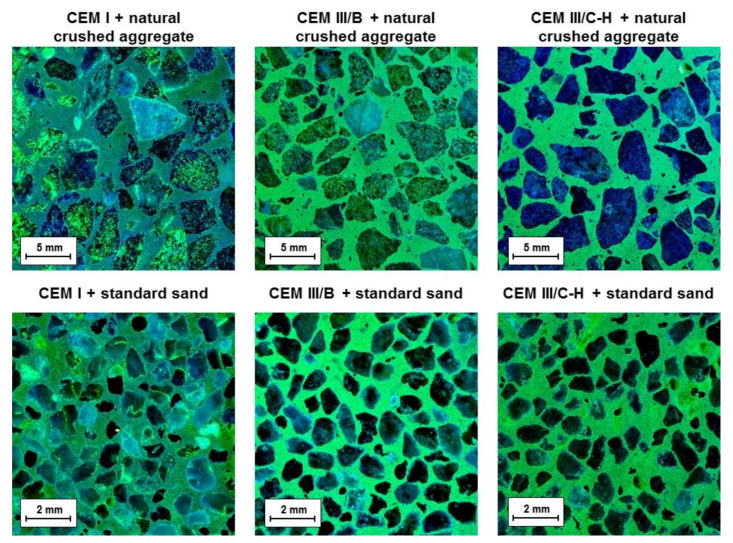
Fracture areas of mortar bars after dye treatment with a uranyl acetate solution under UV light.

**Figure 4 materials-13-03646-f004:**
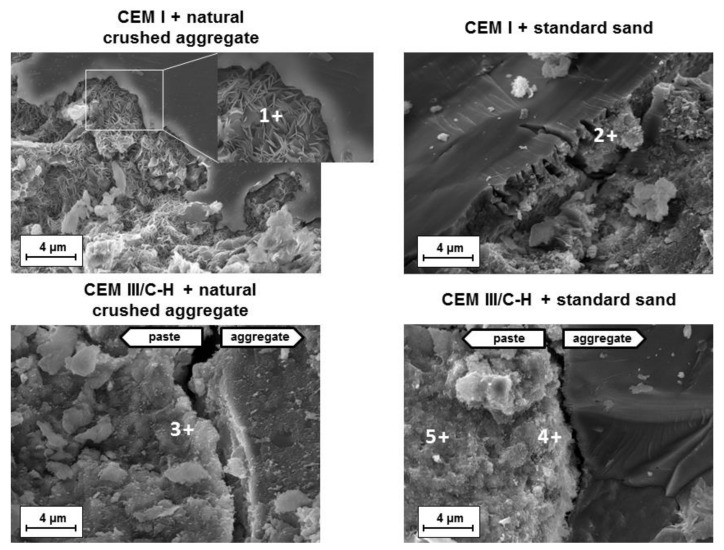
Microstructure of fracture areas of mortar samples after the accelerated expansion test according to ASTM C1567-13.

**Table 1 materials-13-03646-t001:** Chemical compositions of used types of cement as determined by X-ray fluorescence spectrometry.

Cement	Chemical Composition/wt. %										
	SiO_2_	Al_2_O_3_	CaO	Na_2_O	K_2_O	MgO	SO_3_	Fe_2_O_3_	TiO_2_	MnO	LOI
CEM I	20.0	4.5	62.9	0.1	1.3	1.4	3.4	3.2	0.3	0.3	3.4
CEM III/B	31.6	7.4	45.6	0.3	0.8	5.8	3.3	1.4	0.4	0.6	0.4
CEM III/C-H	42.0	7.6	36.4	1.4	0.7	8.6	1.1	0.4	0.8	0.5	1.7

**Table 2 materials-13-03646-t002:** Chemical composition of selected areas determined by EDS.

Place	Mortars	Elemental Composition/at. %									
		O	Na	Mg	Al	Si	Ca	K	Fe	Ca/Si	Al/Ca
1^1^	CEM I + CA	62.3	3.9	0.9	1.9	20.5	9.2	–	1.3	0.44	0.21
2	CEM I + SS	72.0	8.2	1.0	0.9	8.3	9.3	0.1	0.2	1.12	0.10
3	CEM III/C-H + CA	73.1	1.5	2.0	1.7	9.1	12.2	0.3	0.1	1.34	0.14
4	CEM III/C-H + SS	68.2	0.4	0.2	1.3	12.0	17.3	0.4	0.2	1.44	0.08
5	CEM III/C-H + SS	60.7	3.2	2.5	2.2	12.9	17.9	0.5	0.1	1.39	0.12

^1^ The acceleration voltage was decreased to 5 kV.

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
