# Peer review of "Blastfurnace Hybrid Cement with Waste Water Glass Activator: Alkali–Silica Reaction Study"

_materials, 2020, doi:10.3390/ma13163646_

Round 1
Reviewer 1 Report
This paper considers the supplementary cementitious materials as supplementary to Portland cement binding system.
Generally, this paper refers to alkali-silica reaction (ASR) mechanism.
In my opinion work in the presented form exhibits rather valuable impact, nevertheless, I recommend:
1) please add XRD all tested materials before and after hydration,
2) there in no scientific consideration of ASR supported by the relevant scientific techniques, e.g. FT-IR, DTA, Raman etc.
Author Response
This paper considers the supplementary cementitious materials as supplementary to Portland cement binding system.Generally, this paper refers to alkali-silica reaction (ASR) mechanism. In my opinion work in the presented form exhibits rather valuable impact, nevertheless, I recommend:
Dear reviewer,
Thank you for your response. We tried to answer all your comments.
1) please add XRD all tested materials before and after hydration,
XRD of used raw materials was added into the section “Materials and Methods”. However, in our opinion, the usage of XRD method for further investigation makes is not reasonable and possible because it would not provide relevant information. After the hydration process, the observed ASR gel is predominantly amorphous. The identification of rosette‑type crystal phase via XRD is also not reasonable due to strong diffractions of main crystal phases in mortar samples that overlap the phases with very minor content.
2) there in no scientific consideration of ASR supported by the relevant scientific techniques, e.g. FT-IR, DTA, Raman etc.
We believe that our used methods are sufficient for ASR monitoring in hybrid systems. The aim of our paper was, in particular, to verify the resistance to ASR expansion in a hybrid system with high alkali content. Nevertheless, we agree that for further investigation of ASR products the recommended methods are very suitable.
Reviewer 2 Report
The paper by Lukáš Kalina et al. entitled “Blastfurnace Hybrid Cement with Waste Water Glass Activator: Alkali—Silica reaction study” is an interesting study on blastfurnace hybrid cement. The research question is important and the study is original. This article evaluated the potential risk of ASR in high alkaline blastfurnace hybrid cement. The authors report that the beneficial effect of alkali content on the hydration process was also demonstrated. The experiments are properly conducted and clearly presented. The methods section is clear. They concluded that the ASR products were not detected in contrast with mortars based on CEM I cement and the amount of SCMs content seems to be a key parameter for ASR development. They maintain that the possibility of utilization of a new sustainable type of cement with very low Portland clinker content. The discussion section is plausible and well written. The conclusions are sound. This paper reports a number of significant findings that will be of interest to readers of this journal.
Author Response
The paper by Lukáš Kalina et al. entitled “Blastfurnace Hybrid Cement with Waste Water Glass Activator: Alkali—Silica reaction study” is an interesting study on blastfurnace hybrid cement. The research question is important and the study is original. This article evaluated the potential risk of ASR in high alkaline blastfurnace hybrid cement. The authors report that the beneficial effect of alkali content on the hydration process was also demonstrated. The experiments are properly conducted and clearly presented. The methods section is clear. They concluded that the ASR products were not detected in contrast with mortars based on CEM I cement and the amount of SCMs content seems to be a key parameter for ASR development. They maintain that the possibility of utilization of a new sustainable type of cement with very low Portland clinker content. The discussion section is plausible and well written. The conclusions are sound. This paper reports a number of significant findings that will be of interest to readers of this journal.
Dear reviewer,
Thank you for your positive response to our study. Such evaluation motivates us to further work.
Reviewer 3 Report
The paper investigates the alkali silica reaction (ASR) of one-part alkali activated materials prepared with a new type of cement (CEM III/C-H) containing a high amount of blast furnace slag (BFS) and waterglass waste and two traditional cements, namely CEM I and CEM III/B. Mortars were prepared with two types of aggregates: a siliceous sand (SS) and natural crushed aggregates (CA).
Test performed show that mortars prepared with BSF-containing cements resist much better than CEM I to ASR both if prepared with SS and CA. On the contrary, the expansion occurred in mortars manufactured with CEM I leads to a decrease in compressive strength.
In order to improve the paper, some minor revisions are required.
Introduction:
Lines 58-60: the sentence is not clear.
Authors can add something about the sustainability and benefits of using waste waterglass instead of pure waterglass.
Materials and methods:
Mix design of mortars is missing.
Results and discussion:
Why does SS aggregate increase the expansion of mortars compared to CA aggregate?
Why do mortars with CA aggregate show a lower strength than those with SS aggregate?
Why does the compressive strength of mortars manufactured with cements containing slag continue to increase even after immersion in 1N NaOH at 80 °C?
Conclusions:
In the first bullet point, authors can say that ASR does not affect negatively the compressive strength which continues to increase even after the exposure to 1N NaOH at 80 °C.
Author Response
The paper investigates the alkali silica reaction (ASR) of one-part alkali activated materials prepared with a new type of cement (CEM III/C-H) containing a high amount of blast furnace slag (BFS) and waterglass waste and two traditional cements, namely CEM I and CEM III/B. Mortars were prepared with two types of aggregates: a siliceous sand (SS) and natural crushed aggregates (CA). Test performed show that mortars prepared with BSF-containing cements resist much better than CEM I to ASR both if prepared with SS and CA. On the contrary, the expansion occurred in mortars manufactured with CEM I leads to a decrease in compressive strength.
In order to improve the paper, some minor revisions are required.
Dear reviewer,
Thank you for your response. We tried to answer all your comments.
Introduction:
Lines 58-60: the sentence is not clear.
The sentence was improved for better comprehensibility.
Authors can add something about the sustainability and benefits of using waste waterglass instead of pure waterglass.
The explanation of water glass waste usage was added at lines 84-86.
Materials and methods:
Mix design of mortars is missing.
Mix design was always carried out in accordance with the used mentioned standards. In other words, the mix design for mortar samples was different both for expansion tests based on ASTM C1567‑13 and compressive strength testing (EN 196-1). Preparation of mortar samples for mechanical testing was in accordance with EN 196-1 which was added at line 115.
Results and discussion:
Why does SS aggregate increase the expansion of mortars compared to CA aggregate?
The explanation is summarized at lines 165-169.
Why do mortars with CA aggregate show a lower strength than those with SS aggregate?
The explanation is summarized at lines 180-186.
Why does the compressive strength of mortars manufactured with cements containing slag continue to increase even after immersion in 1N NaOH at 80 °C?
The explanation is summarized at lines 174-176.
Conclusions:
In the first bullet point, authors can say that ASR does not affect negatively the compressive strength which continues to increase even after the exposure to 1N NaOH at 80 °C.
The first bullet point was extended as recommended.
Reviewer 4 Report
The herein study, i.e." Blastfurnace Hybrid Cement with Waste Water Glass Activator: Alkali-Silica reaction study" submitted to the Journal Materials, is relevant for the current body of knowledge of the scientific domain related to the alkali-silica reaction of cement.
The article is interesting, and it is written clearly.
However, the author may make use of the following comments:
The standards that are referenced in the text must be listed.
In my opinion, some data is missing in terms of the production of samples and the number of samples for each type of cement.
Author Response
The herein study, i.e." Blastfurnace Hybrid Cement with Waste Water Glass Activator: Alkali-Silica reaction study" submitted to the Journal Materials, is relevant for the current body of knowledge of the scientific domain related to the alkali-silica reaction of cement.
The article is interesting, and it is written clearly.
However, the author may make use of the following comments:
Dear reviewer,
Thank you for your response. We tried to answer all your comments.
The standards that are referenced in the text must be listed.
There is no problem to add the used standards into the reference list. However, the “Instruction for Authors” for Materials journal does not mentioned citation of standards. We can state that our other articles in this journal did not contain a reference to standards as well.
In my opinion, some data is missing in terms of the production of samples and the number of samples for each type of cement.
Mix design was always carried out in accordance with the used mentioned standards including the recommended number of samples. In other words, the preparation of mortar samples was different both for expansion tests based on ASTM C1567‑13 and compressive strength testing (EN 196-1).
Reviewer 5 Report
The manuscript studies the alkali silica reaction on hybrid cements by expansion test, mechanical strength, microstructural analysis and uranyl acetate fluorescence method.
I think that there are two important points that should be revised:
1.- If the presence of BFS produces a reaction with the uranyl cations, the proposed method is not suitable, as the fluorescence around the aggregate will not be detected, as the reaction will always be observed in the binder phase containing BFS.
2.- The main effect is not from the hybrid cement, similar behaviour is observed for the CEMIII. The effect of both aggregates should be also discussed.
There are some parts of the manuscript that can be revised:
3.- Characteristics of the cement samples should be indicated, water/cement ratio; binder/aggregate ratio. Have been modified the previous ratios for the samples with waterglass?. The samples were cured before tested? The same curing conditions were used for all the samples? For example, Cement samples require 25ºC and 70% R.H., for 28 days; what curing conditions were used for the CEM III/C-H samples with only 5% of cement?
4.- What is the chemical composition of the natural crushed aggregate? An also what is the size of both aggregates? Both information are very important in order to understand ASR.
5.- In the paragraph 133-151 only the effect of the cement type is discussed, however the influence of the aggregate should be also studied, since it is clear that until 8 days expansion is lower for all the cements with CA aggregate.
6.- Which samples have been exposed to the uranyl acetate method, i.e. how long have they been cured and what curing conditions have been used?
7.- What is the age of the samples of SEM? SEM images of the CEM I + aggregate is the image of plates of portlandite, not C-S-H.
8.- SEM image of the CEM III/CH shows not good adhesion between aggregate and the binder, so how can the author explain this?
9.- Conclusion 4 is not really studied in the manuscript since no mineralogical composition from XRD or FTIR is determined.
Author Response
The manuscript studies the alkali silica reaction on hybrid cements by expansion test, mechanical strength, microstructural analysis and uranyl acetate fluorescence method.
I think that there are two important points that should be revised:
1.- If the presence of BFS produces a reaction with the uranyl cations, the proposed method is not suitable, as the fluorescence around the aggregate will not be detected, as the reaction will always be observed in the binder phase containing BFS.
It is true that the detection of potential fluorescence around aggregate in the fluorescing matrix can be an issue. Nevertheless, if you look at the fracture areas of mortar bars with CEM I type of cement, you can see ASR products both around and across the aggregate particles. This phenomenon is not detected in the case of mortars with CEM III/B and CEM III/C-H types of cement. Moreover, the results from all other adopted methods do not show evidence of deleterious ASR gel and therefore we do not expect its formation around aggregate grains in both CEM III cements.
2.- The main effect is not from the hybrid cement, similar behaviour is observed for the CEMIII. The effect of both aggregates should be also discussed.
Yes, we agree with this statement. The aim of our paper was, in particular, to verify the resistance to ASR in a hybrid system with high alkali content and comparison with other types of cement and CEM III type was selected as the most similar to our hybrid cement as possible. The discussion about the effect of both aggregates was added at lines 165-169 and 180-186.
There are some parts of the manuscript that can be revised:
3.- Characteristics of the cement samples should be indicated, water/cement ratio; binder/aggregate ratio. Have been modified the previous ratios for the samples with waterglass?. The samples were cured before tested? The same curing conditions were used for all the samples? For example, Cement samples require 25ºC and 70% R.H., for 28 days; what curing conditions were used for the CEM III/C-H samples with only 5% of cement?
Water/cement ratio as well as binder/aggregate ratio is given by used and mentioned standards (ASTM C1567‑13; EN 196‑1). Water/cement ratio was not modified for samples with WG-waste because this by-product material was mixed in a dry state (line 95) with clinker and slag and milled together to produce the hybrid cement and therefore the whole hybrid cement dose was calculated as a cement or binder in the mentioned ratios. The curing processes were the same for all samples during specific analysis. All curing conditions were given by the requirement of used standard methods.
4.- What is the chemical composition of the natural crushed aggregate? An also what is the size of both aggregates? Both information are very important in order to understand ASR.
The mineralogical composition of used aggregates was added into the section “Materials and Methods”. The particle size distribution for both aggregates was adjusted according to the used standard (see ASTM C1567‑13).
5.- In the paragraph 133-151 only the effect of the cement type is discussed, however the influence of the aggregate should be also studied, since it is clear that until 8 days expansion is lower for all the cements with CA aggregate.
The explanation is added at lines 165-169.
6.- Which samples have been exposed to the uranyl acetate method, i.e. how long have they been cured and what curing conditions have been used?
The samples were identified after the ASTM C 1567‑13 procedure was applied. It is written at lines 131-132.
7.- What is the age of the samples of SEM? SEM images of the CEM I + aggregate is the image of plates of portlandite, not C-S-H.
The samples were identified after the ASTM C 1567‑13 procedure as in previous case. In the first image of mortar with CEM I is definitely not portlandite plates. See the chemical composition in Table 2 place 1. In order to decrease the interaction volume as much as possible, we reduced the accelerating voltage. The similar plates (rosettes) were also identified in previous study:
Cyr, M.; Pouhet, R. Resistance to alkali-aggregate reaction (AAR) of alkali-activated cement-based binders. In Handbook of alkali-activated cements, mortars and concretes, Eds. Elsevier: 2015
8.- SEM image of the CEM III/CH shows not good adhesion between aggregate and the binder, so how can the author explain this?
This is only just about the place selection. We tried to find the places where the interphase between aggregate and binder phase will be well recognized.
9.- Conclusion 4 is not really studied in the manuscript since no mineralogical composition from XRD or FTIR is determined.
This conclusion was revised to fit better the obtained results. (Former statement was rather based on the findings from other papers and EDS data.).
Round 2
Reviewer 4 Report
I think the authors should make the requested changes.
Reviewer 5 Report
I agree with all the comments made in the paper and accept its publication.